# PRDX2 Knockdown Inhibits Extracellular Matrix Synthesis of Chondrocytes by Inhibiting Wnt5a/YAP1/CTGF and Activating IL-6/JAK2/STAT3 Pathways in Deer Antler

**DOI:** 10.3390/ijms23095232

**Published:** 2022-05-07

**Authors:** Xuyang Sun, Xiaoying Gu, Jingna Peng, Liguo Yang, Xinxin Zhang, Zaohong Ran, Jiajun Xiong

**Affiliations:** Key Lab of Agricultural Animal Genetics, Breeding and Reproduction of Ministry of Education, College of Animal Science and Technology, Huazhong Agricultural University, Wuhan 430070, China; sunxuyangabc@163.com (X.S.); gxy19970310@163.com (X.G.); peng980906@163.com (J.P.); yangliguo2006@foxmail.com (L.Y.); zxx1144795936@163.com (X.Z.); ranzaohong@webmail.hzau.edu.cn (Z.R.)

**Keywords:** PRDX2, antler chondrocytes, CTGF, IL-6, cartilage extracellular matrix

## Abstract

Although peroxiredoxin 2 (PRDX2) plays a vital role in relieving oxidative stress, its physiological function in cartilage development remains almost unknown. In this study, we found that the expression of PRDX2 significantly increased in the chondrocytes compared with pre-chondrocytes. PRDX2 knockdown significantly decreased the expression of extracellular matrix (ECM) protein (Col2a and Aggrecan), which led to blocked cartilage formation. Moreover, PRDX2 knockdown also inhibited the expression of connective tissue growth factor (CTGF). CTGF is an important growth factor that regulates synthesis of ECM proteins. We explored the possible regulatory mechanism by which PRDX2 regulated the expression of CTGF. Our results demonstrated that PRDX2 knockdown downregulated the expression of CTGF by inhibiting Wnt5a/Yes-associated protein 1 (YAP1) pathway. In addition, PRDX2 knockdown promoted the expression of interleukin 6 (IL-6), indicating PRDX2 expression had an anti-inflammatory function during antler growth. Mechanistically, PRDX2 knockdown promoted cartilage matrix degradation by activating the IL-6-mediated Janus Kinase 2/Signal Transducer and Activator of Transcription 3 (JAK2/STAT3) signaling pathway. These results reveal that PRDX2 is a potential regulator that promotes cartilage extracellular matrix synthesis.

## 1. Introduction

Deer antler, a bony organ, grows in vitro, is easy to observe and can periodically regenerate. Antler cartilage is formed by endochondral ossification [1]. During this period, the tip of the antler tissue is stored through a transformation of perichondrium into mesenchymal cells, which differentiate to pre-chondrocytes and finally transform into chondrocytes (Figure 1) [2,3]. During early chondrogenesis, chondrocytes continue to divide and specifically express ECM proteins including type II collagen alpha (Col2a), cartilage oligomeric matrix protein (COMP) and Aggrecan, which promote cartilage ECM formation [4]. In the meantime, parts of chondrocytes step down from the cell cycle and continue to differentiate into hypertrophic chondrocytes. In hypertrophic chondrocytes (during late chondrogenesis), Runt-related transcription factor 2 (Runx2) is dominantly expressed and regulates type X collagen alpha (Col10a) expression [5]. Runx2 interacts with Osterix to induce the expression of matrix metalloproteinases 13 (MMP13), which results in calcification of matrices [6].

The ECM, the living environment of chondrocytes, is a complex network composed of collagen, chondroitin sulfate, proteoglycans and various signaling molecules. It provides chondrocytes with a three-dimensional pore structure to facilitate their attachment, proliferation and secretion of the matrix [7,8]. Once the cartilage extracellular matrix is degraded, it can cause cartilage diseases such as osteoarthritis [9]. Therefore, exploring the mechanism of extracellular matrix formation has become an important means to treat cartilage damage repair.

PRDX2 is a member of the peroxiredoxin family which protects cells from oxidative stress by removing H_2_O_2_ and controlling reactive oxygen species (ROS) level [10]. Furthermore, PRDX2 is also involved in mediating other biological functions. It has been reported that PRDX2 as a potential inflammatory mediator regulates inflammatory response [11]. The high expression of PRDX2 is also related to the survival and proliferation of cancer cells, including gastric cancer, lung cancer and colon cancer [12,13]. A recent study has indicated that PRDX2 mediates atherosclerosis progression [14]. This also implies that PRDX2 has a wide range of biological activities in different types of cells. However, the biological function of PRDX2 for cartilage development remains unexplored.

Wnt signaling plays a significant role in various biological processes, such as angiogenesis, tumor growth, immune response and cartilage development [15,16,17,18]. The canonical Wnt pathway relies on the stabilization and nuclear translocation of β-catenin to regulate the expression of downstream genes. The activation of the canonical Wnt signal prevents the degradation of β-catenin by the APC complex, and β-catenin accumulates in the cytoplasm, thus realizing the transfer from the cytoplasm to the nucleus [19]. The non-canonical Wnt signaling pathway has multiple regulatory mechanisms, including the Wnt/mTOR, Wnt/JNK, Wnt/Ca^+^ and Wnt/YAP pathways, playing important roles in regulating cell growth, differentiation and apoptosis [20,21,22,23].

In recent years, it has been discovered that the Wnt/YAP1 signaling pathway can mediate various biological effects. YAP acts as a downstream effector of Wnt proteins to alter the Wnt signaling pathway, in which Wnt5a/b can induce the activation of YAP/TAZ signaling [24]. YAP1, a nuclear transcription factor and the key mediator of the Hippo signaling pathway, plays an important role in regulating cell proliferation, differentiation and survival [25,26]. Previous studies have shown that YAP1 can bind to TEADs in the nucleus and act as a transcriptional co-activator to regulate the expression of target genes, such as CYR61 and CTGF [27,28].

The CTGF/CCN2 (connective tissue growth factor) is the second member of the CCN family implicated in cell proliferation, differentiation and ECM production [29]. Many in vitro and in vivo studies have shown that CTGF as an osteogenesis-related protein plays an important role in regulating endochondral ossification [30]. CTGF promotes new bone formation by regulating extracellular matrix accumulation and intramembranous osteogenesis [31]. Loss of CTGF can cause severe chondrodysplasia [32]. In addition, IL-6 is a key inflammatory factor that regulates immune and inflammatory responses [33]. Previous studies have shown that inflammation is a common cause of cartilage destruction. The IL-6-mediated JAK2/STAT3 signaling pathway can induce chondrocyte apoptosis and MMP expression [34,35].

In this study, we analyzed the expression levels of PRDX family members in pre-chondrocytes and chondrocytes and found that as antler chondrocytes matured, PRDX2 expression increased significantly. It is unclear whether PRDX2 is involved in the process of cartilage development. Here, we aimed to investigate the specific biological effect of PRDX2 during the rapid growth of deer antler and explored the potential regulatory mechanism.

## 2. Results

### 2.1. Antler Chondrocytes Can Highly Express a Variety of Peroxiredoxins, and the Expression of PRDX2 Upregulates as the Chondrocytes Mature

Although the growth rate of an antler is extremely quick, the consumption of large amounts of oxygen does not cause oxidative stress. We analyzed the expression level of the PRDX family protein with qRT-PCR in antler chondrocytes. We found that PRDX1-6 were expressed in the antler chondrocytes, and the expression levels of PRDX1 and PRDX2 were high (Figure 2A). In a previous study, we successfully isolated and identified antler pre-chondrocytes and chondrocytes [36]. Here, we detected the expression levels of PRDXs between the pre-chondrocytes and the chondrocytes. Chondrocytes can highly express Col2a (chondrocytes marker molecule) compared to pre-chondrocytes (Figure 2B). We found that the expression of PRDX2 significantly increased as the chondrocytes matured (Figure 2C). We detected the location and expression of PRDX2 in the pre-cartilage layer and cartilage layer with immunohistochemistry staining. PRDX2 protein showed a strong signal in the cartilage layer of deer antler (Figure 2D).

### 2.2. PRDX2 Knockdown Inhibits the Synthesis of Cartilage Matrix Proteins

PRDX2 expression significantly increased in the chondrocytes compared with pre-chondrocytes, and we proposed that it may mediate early chondrogenesis. To determine whether PRDX2 mediates chondrogenesis, the interference fragment targeting the PRDX2 gene was transfected into proliferating chondrocytes of deer antler. The interference efficiency of PRDX2 was detected using Western blot analysis and qRT-PCR (Figure 3A,B). PRDX2 knockdown significantly inhibited the expression of cartilage matrix proteins (Col2a, Aggrecan and COMP) compared with the negative control (Figure 3C). In addition, the results from toluidine blue staining and Alcian blue staining showed PRDX2 knockdown reduced the synthesis of glycosaminoglycan, which inhibited the formation of the cartilage extracellular matrix (Figure 3D).

### 2.3. PRDX2 Knockdown Inhibits the Expression of CTGF

In vivo and in vitro studies have shown that CTGF plays a vital role in regulating the synthesis of the ECM and maintaining chondrocyte phenotype. We investigated whether PRDX2 regulated the synthesis of cartilage matrix proteins through the transcription of CTGF. We found that PRDX2 knockdown significantly inhibited the expression of CTGF (Figure 4A,B).

### 2.4. PRDX2 Knockdown Inhibits the Expression of Wnt5a and Promotes Nuclear Translocation of β-Catenin 

The Wnt/β-catenin pathway is a key regulator during chondrogenesis. To further explore how PRDX2 regulated the expression of CTGF, we evaluated whether PRDX2 regulated the expression of CTGF though activating the Wnt/β-catenin signaling pathway. We found that PRDX2 knockdown significantly decreased Wnt5a expression but increased β-catenin activity (Figure 5A). The immunofluorescence results showed that after PRDX2 knockdown, β-catenin had a strong signal in the nucleus (Figure 5B). In addition, PRDX2 knockdown significantly inhibited phosphorylation of GSK-3β and enhanced GSK-3β activation (Figure 5C,D). These results indicate that PRDX2 is involved in Wnt5a pathways (independent of β-catenin).

### 2.5. PRDX2 Knockdown Reduces the Expression of CTGF by Inhibiting the Activity of YAP1

Previous studies have indicated that Wnt5a can induce YAP/TAZ activation to change the canonical Wnt signaling pathway [24,37]. Here, we performed Western blot analysis to examine the transcription activity of YAP1 after PRDX2 knockdown. Treatment with siPRDX2 significantly increased the expression of p-YAP1 and decreased the nuclear translocation of YAP1 (Figure 6A). The immunofluorescence results also showed that after PRDX2 knockdown, YAP1 had a weak signal in the nucleus compared with the negative control (Figure 6B). Next, the interference fragment targeting the YAP1 gene was transfected into chondrocytes. The protein level of YAP1 was significantly reduced compared with the negative control (Figure 6C,E). The expression of CTGF was significantly reduced after YAP1 knockdown by q-PCR (Figure 6D). These results indicate that PRDX2 regulated the expression of CTGF depending on the transcriptional activity of YAP1.

### 2.6. PRDX2 Knockdown Activates IL6-Induced JAK2/STAT3 Signaling Pathway in Chondrocytes

The JAK2/STAT3 pathway plays an important role in osteoarthritis (OA) pathogenesis. Activation of the JAK2/STAT3 pathway can accelerate the degradation of the cartilage matrix. Figure 7A shows that PRDX2 knockdown promotes the expression of IL-6. As expected, PRDX2 knockdown upregulated the phosphorylation levels of JAK2 and STAT3 (Figure 7B,C). These results indicate PRDX2 knockdown induced inflammatory response and inhibited the expression of cartilage matrix proteins through activating the IL6-induced JAK2/STAT3 signaling pathway.

## 3. Discussion

PRDX2 is a very powerful ROS-scavenging protein compared with other PRDX members involved in important physiological functions. In this work, we found that PRDX2 was expressed at different stages of chondrogenesis. PRDX2 expression was significantly increased in chondrocytes compared with pre-chondrocytes. PRDX2 knockdown inhibited the expression of Col2a, Aggrecan, COMP and reduced the synthesis of glycosaminoglycan, indicating that PRDX2 was involved in the cartilage ECM formation and maintained chondrocyte phenotype. Furthermore, PRDX2 knockdown markedly increased the expression of IL-6, indicating PRDX2 expression had an anti-inflammatory function during antler growth.

Previous studies have shown that CTGF is a significant matricellular protein that regulates cartilage growth [38,39,40]. Moreover, exogenous CTGF treatment increases the expression of cartilaginous matrix proteins, such as Col2a and Aggrecan, and promotes chondrocytes proliferation [41]. Here, we found that PRDX2 knockdown inhibited the expression of CTGF. PRDX2 involvement in regulating the cartilage matrix protein expression may depend on the activity of CTGF. We further explored the relationship between PRDX2 and CTGF as well as possible regulatory mechanisms. Previous studies have shown that YAP1 can bind to TEADs in the nucleus and act as a transcriptional co-activator to regulate the expression of target genes, such as CYR61 and CTGF [27,28]. We suspect that PRDX2 may affect the transcription of YAP1 in antler chondrocytes. Here, we observed that PRDX2 knockdown significantly promoted YAP1 phosphorylation and inactivation. In addition, relevant studies have shown that inhibition or downregulation of GSK-3β can promote nuclear aggregation of YAP1 [42]. PRDX2 knockdown upregulated the expression of GSK-3β (Figure 4C). These results suggested that PRDX2 regulated the expression of CTGF by activating the YAP1 signaling pathway. However, it is unclear how PRDX2 is associated with YAP1, and further study is required to explore this.

Non-canonical Wnt signaling mediates a variety of physiological functions, including embryonic development, cell differentiation and inhibition of canonical Wnt signaling [43,44]. Wnt5a has traditionally been considered a common ligand in non-canonical Wnt signaling. Wnt5a-driven signaling is able to inhibit canonical Wnt signaling by inhibiting the expression of Wnt target genes [24]. Ror2 receptor, as a major receptor of Wnt5a, plays a vital role in mediating non-canonical Wnt signaling. Wnt5a can inhibit the transcriptional activity of β-catenin by activating the phosphorylation of the Ror2 receptor [45,46]. We found that PRDX2 knockdown inhibited the Wnt5a expression and increased the β-catenin accumulation (Figure 4A). Obviously, Wnt5a inhibited the transcriptional activity of β-catenin. In addition, related research results show that Wnt5a can act as an activator of YAP/TAZ signaling, thereby altering the Wnt signaling pathway [24]. Our results indicated that PRDX2 activated the YAP/TAZ signal through Wnt5a in the chondrocytes of deer antler.

CTGF has been identified a vital transcription factor that promotes synthesis of cartilage matrix proteins including Col2a and Aggrecan [41]. Degradation of the extracellular matrix has long been a hallmark of arthritic diseases, and inflammatory responses are the main cause of a series of structural damage [47]. A large number of inflammatory mediators including IL-6, TNF-α and IL-1β were detected in osteoarthritis [48]. In vitro and in vivo studies have shown that the JAK2/STAT3 pathway plays vital roles in osteoarthritis (OA) [35,49]. The activation of the JAK2/STAT3 pathway resulted in the production of matrix metalloproteinases (MMPs), inducible nitric oxide synthase (iNOS) and cyclooxygenase-2, which would lead to cartilage destruction [34]. IL-6 is a well-known target protein upstream of the JAK2/STAT3 signaling pathway, and it can regulate articular cartilage degradation in OA [34,50]. PRDX2 knockdown inhibited the expression of cartilage matrix proteins by activating the IL6-induced JAK2/STAT3 signaling pathway in chondrocytes.

It is well known that cartilage disruption and loss of the extracellular matrix are the most distinctive features of OA [51]. Counteracting cartilage degradation by stimulating chondrocyte proliferation and extracellular matrix synthesis is a potential treatment for OA. In addition, the results of a study showed that PRDX2 expression showed lower levels in the cartilage tissue of OA patients [52]. In this study we found that PRDX2 knockdown downregulated the expression of cartilage matrix proteins. We reported that PRDX2 regulated CTGF expression through the Wnt5a/YAP1 pathway. Our data indicated that PRDX2 promoted extracellular matrix formation by inhibiting the IL-6/JAK2/STAT3 pathway. However, we only explored the effect of cartilage matrix protein expression by knocking down the expression of PRDX2. Whether overexpression of PRDX2 can promote the formation of the cartilage ECM is unclear. In addition, whether PRDX2 expression regulates chondrocyte differentiation remains to be further investigated.

## 4. Materials and Methods

### 4.1. Antler Issue Collection and Cell Culture

Deer antler samples (four antlers) were collected (Jinsanxin Farm, Wuhan, China). These *cervus nippon* are native to the northeastern part of China. The tips of antler tissues (about 5 cm) about 60 days after casting from healthy deer (two years old) were dissected into different layers (mesenchyme layer, pre-cartilage, and cartilage layer) separately as described previously [2]. Dissected tissues were cut up in DMEM/high glucose medium (Hyclone, GE Healthcare, Logan, UT, USA), centrifuged at 1000 rpm for 2 min and using 0.2% collagenase II (Sigma-Aldrich, Marlborough, MA, USA) digested for 30 min. The digested tissue layers were filtered through a cell strainer (pore size: 100 μm), and the cell suspension was centrifuged at 1300 rpm for 4 min. The cell pellets were resuspended in DMEM/high glucose medium containing 10% FBS (fetal bovine serum). Antler cells were cultured at 37 °C with 5% CO_2_. Finally, cell cryopreservation solution was added to freeze the cells in liquid nitrogen for long-term storage.

### 4.2. Immunohistochemistry Staining

For immunohistochemistry staining, antigen-treated, paraffin-embedded tissue was exposed to 3% hydrogen peroxide in the dark, and decolonization was performed with PBS 3 times for 5 min each. Blocking of the paraffin section was performed using PBS with goat serum for 40 min, and it was incubated with PRDX2 antibody (1:500) (Abcam, ab109367) at 4 °C overnight. Subsequently, the paraffin sections were washed with 1X-PBS 3 times and incubated with goat anti-rabbit IgG at 37 °C for 30 min following PBS washing. Before sealing paraffin sections with a neutral stain, they were counterstained using hematoxylin for 2 min and subjected to ddH_2_O washing and 1% hydrochloric acid alcohol differentiation for a few seconds, rinsed with ddH_2_O until ammonia water returned to blue and rinsed with ddH_2_O again.

### 4.3. RNA Interference

Antler chondrocytes were seeded in cell culture plates up to 60–70% confluence. The cells were transfected with 100 nM siRNA by Lipofectamine RNAiMAX Reagent in Opti-MEM medium (Life Technologies, Inc., Carlsbad, CA, USA) according to the instructions as compared to the control. After 48 h of transfection, cells were harvested for mRNA expression or protein expression. The siRNA sequence used in this study is as follows. PRDX2: 5′- AGGAAUAUUUCUCCAAACATT -3′, YAP1: 5′- GGUGACACUAUCAACCAAATT -3′.

### 4.4. Total RNA and Quantitative Real-Time PCR (qRT-PCR)

Post-transfection (48 h), total RNA from Antler cells was extracted using an RNA kit (Cat R6834-02, Omega Bio-Tek, Norcross, GA, USA) according to the kit instructions. Subsequently, the purity of RNA was determined for the absorbance at 260/280. The RNA with the absorbance value of 1.8–2.1 was used with the cDNA first-trans synthesis kit (CatKR118-02, TIANGEN Biotech, Beijing, China) for reverse transcription into cDNA. The expression of genes was analyzed using the method of 2^−ΔΔCT^. All the targeted primers were designed by Primer 5.0 software (Primer Biosoft, Palo Alto, CA, USA) (Table 1). 

### 4.5. Western Blot Assay

Antler cells (10) were collected and lysed for 20 min using lysis buffer (RIPA lysis and 50 × Cocktail) (Servicebio, Wuhan, China). Protein concentration was examined using the BCA protein assay (Servicebio, Wuhan, China). The total protein (20 μg) was separated by SDS-polyacrylamide gel electrophoresis. Subsequently, proteins were transferred to polyvinylidene difluoride membranes and blocked with 5% skimmed milk or bovine serum albumin (BSA) powder with 0.1% Tween-20 in TBS for 2 h and incubated with primary antibody at 4 °C overnight. The following antibodies were used: anti-YAP1 (GTX129151, GeneTex, San Antonin, TX, USA, 1:1000), anti-p-YAP1 (S127) (ab76252, Abcam, Cambridge, UK, 1:2000), anti-GSK3 beta (phosphor S9) (ab75814, Abcam, 1:1000), anti-PRDX2 (ab109367, Abcam, 1:2000), anti-β-catenin (ab32572, Abcam, 1:2000), anti-Wnt5a (ab179824, Abcam, 1:2000), anti-Jak2 (3230, Cell Signaling Technology (CST) Biological reagents Company Ltd., Shanghai, China, 1:1000), anti-p-JAK2 (Try1007/1008) (3771, CST, 1:1000), anti-p-STAT3 (Tyr705) (9145, CST, 1:1000) and anti-Col2a (15943, Proteintech, Wuhan, China, 1:1000). After washing, membranes were incubated with secondary antibodies for 2 h. Lastly, the membranes were incubated with ECL chemiluminescence reagent and exposed to X-ray film for the observation of protein bands. 

### 4.6. Immunofluorescence Assay

Cells were grown on circular glass coverslips in 24-well plates, fixed with 4% paraformaldehyde (15 min) and then permeabilized using TritonX-100 (3 min). After washing with PBS, cell coverslip blocking was performed using BSA 5% for 30 min, and then the coverslips were incubated with primary antibodies at 4 °C overnight. Cell coverslips were washed 3 times with PBST and incubated with FITC or CY3-conjugated secondary antibodies (Servicebio, Wuhan, China). Lastly, counterstaining of the nucleus was performed with DAPI (Servicebio, Wuhan, China).

### 4.7. Toluidine Blue and Alcian Blue Staining

The adherent cells were washed 3 times with phosphate buffered saline solution and fixed with 4% paraformaldehyde at room temperature for 15 min. Then, the cells were incubated at room temperature with the prepared toluidine blue solution (Servicebio, Wuhan, China) for 2 h and washed with distilled water 3 times. Prepared 0.3% Alcian blue solution (Servicebio, Wuhan, China) was incubated at room temperature for 2 h and then washed with distilled water. Finally, an upright microscope was used to collect pictures. 

### 4.8. Statistical Analysis

Analysis of data was performed using GraphPad Prism 5.0 software (GraphPad Software, Inc., San Diego, CA, USA). Data were summarized as mean ±standard deviation (SD), and a significant difference between two groups was determined using the *t* test. *p* < 0.05 indicated a statistically significant difference.

## 5. Conclusions

Our results indicate that PRDX2 has a positive effect on chondrogenesis. PRDX2 expression can effectively promote the production of extracellular matrix proteins. Mechanistically, on the one hand, PRDX2 regulated the expression of CTGF though Wnt5a/YAP1 pathway. On the other hand, PRDX2 suppressed inflammation response by inhibiting IL-6/JAK2/STAT3 pathway in antler chondrocytes, which facilitated the formation of the cartilage extracellular matrix. 

## Figures and Tables

**Figure 1 ijms-23-05232-f001:**
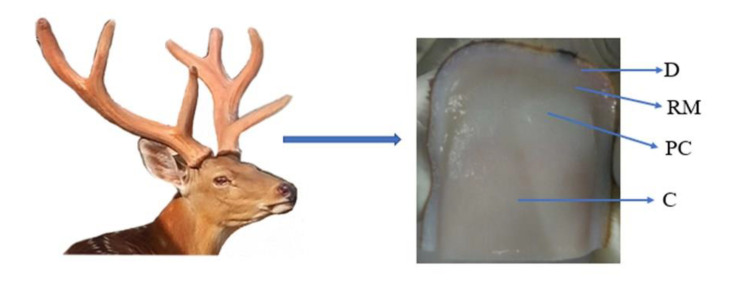
Longitudinal section of antler tip tissue. D, dermis; RM, reserve mesenchymal; PC, pre-cartilage; C, cartilage.

**Figure 2 ijms-23-05232-f002:**
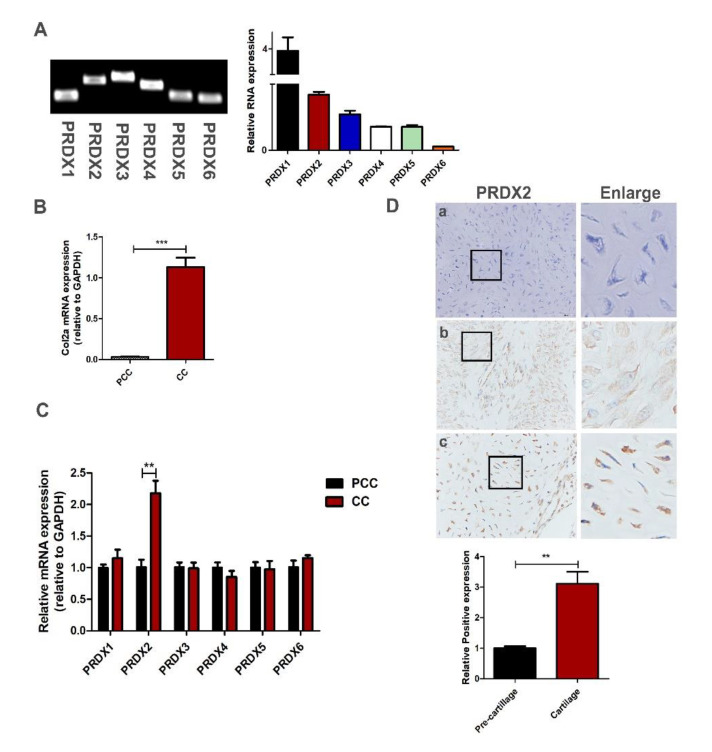
Antler chondrocytes can highly express a variety of peroxiredoxins, and the expression of PRDX2 upregulates as the chondrocytes mature. (**A**) The expression level of PRDX family protein by qRT-PCR in antler chondrocytes. (**B**) The expression level Col2a (chondrocyte marker molecule) by qRT-PCR in antler pre-chondrocytes (PCC) and chondrocytes (CC). (**C**) The expression level of PRDX1-6 by qRT-PCR in PCC and CC. (**D**) Immunohistochemistry staining of PRDX2 in the antler pre-cartilage layer and cartilage layer. (**a**) Negative control, (**b**) pre-cartilage layer, (**c**) cartilage layer. Scale bar, 50 μm. The data include the means ± SD of three independent experiments. ** *p* < 0.01, *** *p* < 0.001.

**Figure 3 ijms-23-05232-f003:**
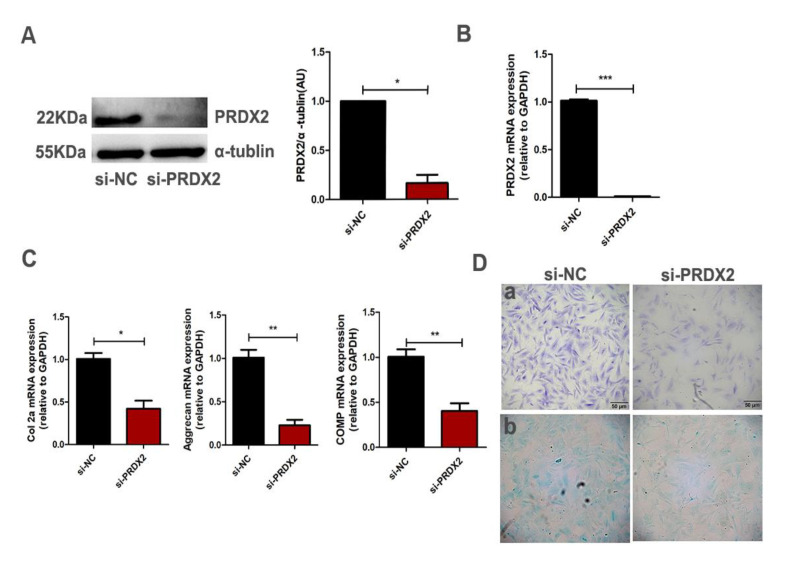
PRDX2 knockdown inhibits the synthesis of cartilage matrix proteins. (**A**) The PRDX2 expression level was decreased after treatment with PRDX2 siRNA in the chondrocytes for 72 h by Western blot. (**B**) The PRDX2 expression level was decreased after treatment with PRDX2 siRNA for 72 h by q-PCR. (**C**) The Col2a, Aggrecan and COMP expression levels were decreased after treatment with PRDX2 siRNA for 72 h by qRT-PCR (**D**) toluidine blue staining (**a**) and Alcian blue staining (**b**) were performed after treatment with PRDX2 siRNA for 72 h. Scale bar, 50 μm. The data include the means ± SD of three independent experiments. * *p* < 0.05, ** *p* < 0.01, *** *p* < 0.001.

**Figure 4 ijms-23-05232-f004:**
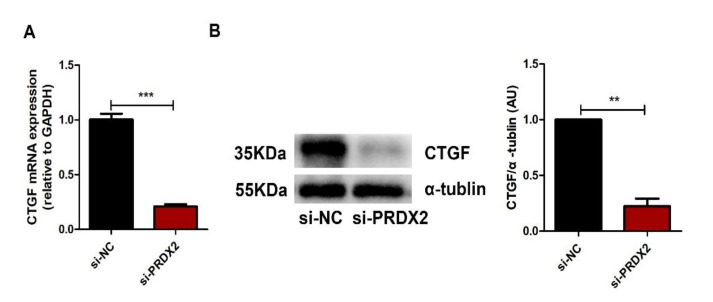
PRDX2 knockdown inhibits the expression of CTGF. (**A**) The CTGF expression level was decreased after treatment with PRDX2 siRNA in the proliferative chondrocytes for 72 h by qPCR. (**B**) The CTGF protein level was detected after treatment with PRDX2 siRNA in the antler chondrocytes for 72 h by Western blot. The data include the means ± SD of three independent experiments. ** *p* < 0.01, *** *p* < 0.001.

**Figure 5 ijms-23-05232-f005:**
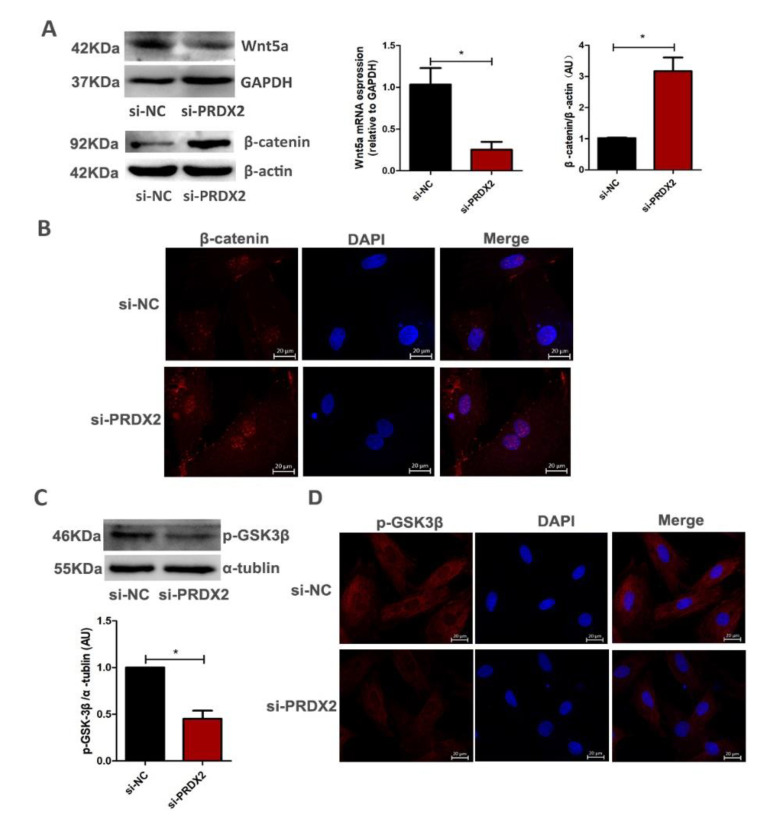
PRDX2 knockdown inhibits the expression of Wnt5a and promotes nuclear translocation of β-catenin. (**A**) The Wnt5a and β-catenin expression levels were decreased after treatment with PRDX2 siRNA in the proliferative chondrocytes for 72 h by Western blot. (**B**) Control or PRDX2 siRNA was transfected for 72 h in proliferative chondrocytes. Localization of β-catenin was determined by immunofluorescence staining. The nuclei were stained with DAPI (blue), and the β-catenin was detected with CY3 (red). Scale bar, 20 μm. (**C**) The p-GSK-3β protein level was detected after treatment with PRDX2 siRNA in the antler chondrocytes for 72 h by Western blot. (**D**) Localization of p-GSK3β was determined by immunofluorescence staining. The nuclei were stained with DAPI (blue), and the p-GSK3β was detected with CY3 (red). Scale bar, 20 μm. The data include the means ± SD of three independent experiments. * *p* < 0.05.

**Figure 6 ijms-23-05232-f006:**
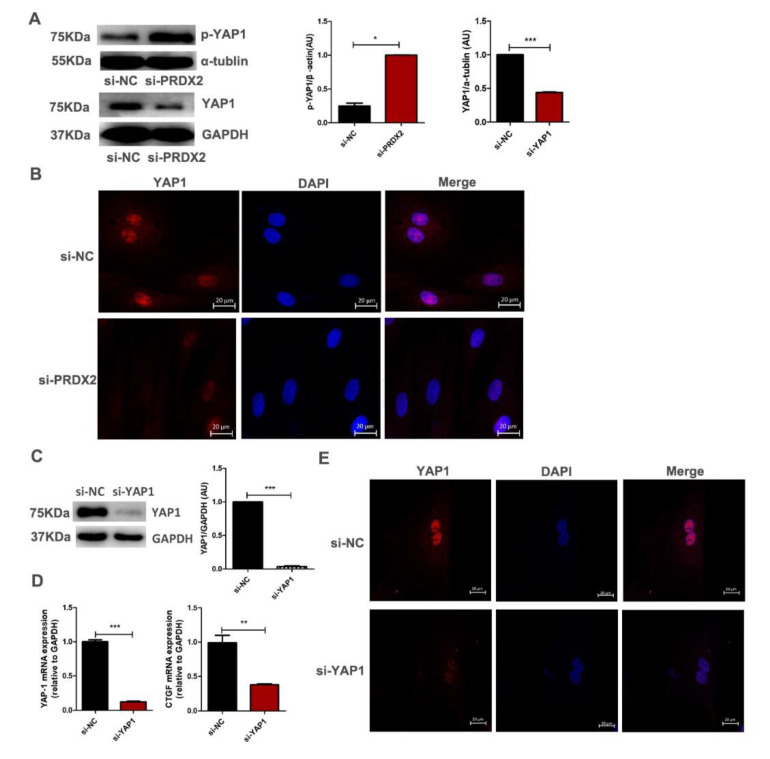
PRDX2 knockdown reduces the expression of CTGF by inhibiting the activity of YAP1. (**A**) The YAP1 and p-YAP1 expression levels were decreased after treatment with PRDX2 siRNA in antler chondrocytes for 72 h by Western blot. (**B**) Control or PRDX2 siRNA was transfected for 72 h. Localization of YAP1 was determined by immunofluorescence staining. The nuclei were stained with DAPI (blue), and the YAP1 was detected with CY3 (red). Scale bar, 20 μm. (**C**) The protein level of YAP1 was decreased after treatment with YAP1 siRNA for 72 h by Western blot. (**D**) The CTGF expression level was decreased after treatment with YAP1 siRNA for 72 h by q-PCR. (**E**) The interference efficiency of YAP1 was examined by immunofluorescence staining. The YAP1 was detected with CY3 (red). Scale bar, 20 μm. The data include the means ± SD of three independent experiments, * *p* < 0.05, ** *p* < 0.01, *** *p* < 0.001.

**Figure 7 ijms-23-05232-f007:**
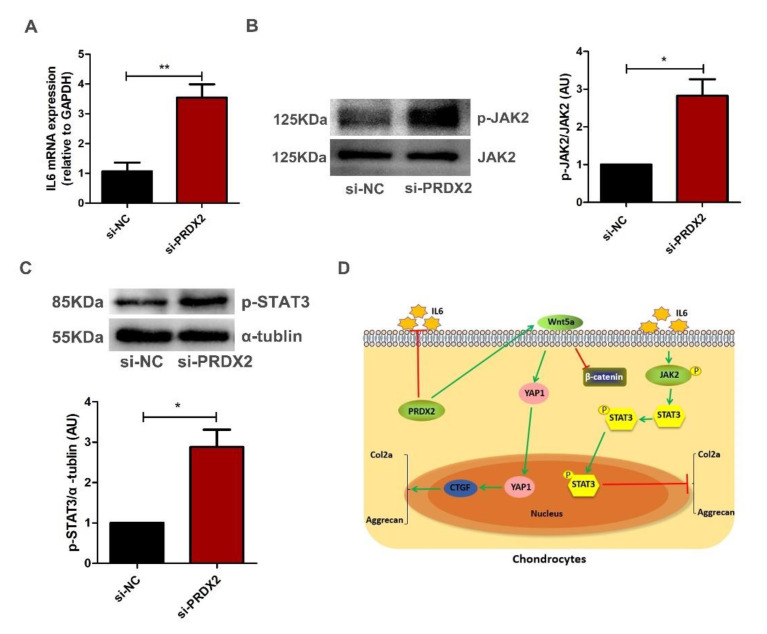
PRDX2 knockdown activates IL6-induced JAK2/STAT3 signaling pathway in chondrocytes. (**A**) IL-6 expression levels were decreased after treatment with PRDX2 siRNA in the hypertrophic chondrocytes for 72 h by qRT-PCR. (**B**,**C**) The p-JAK2 and p-STAT3 protein levels were increased after treatment with PRDX2 siRNA for 72 h by Western blot. (**D**) PRDX2 knockdown inhibits extracellular matrix synthesis of chondrocytes by inhibiting Wnt5a/YAP1/CTGF and activating IL-6/JAK2/STAT3 pathways in deer antler. The data include the means ± SD of three independent experiments, * *p* < 0.05, ** *p* < 0.01.

**Table 1 ijms-23-05232-t001:** Primers used for RT-PCR in this study.

Gene	Primer Sequence (5′-3′)
Col2a	F: GAGGCAGCCGGCAACCTGAG
	R: TGCGAGCTGGGTTCTTGCGG
YAP-1	F: GTTCCAACCAGCAGCAACAG
	R: GGTAACTGGCTACGGAGAGC
PRDX2	F: GCTGAACATTCCCCTGCTG
	R: CGTCCACATTGGGCTTGAT
Aggrecan	F: CAACCTCCTGGGAGTGAGGA
	R: GCTTTGCCGTGAGGATCAC
Wnt5a	F: CTCCTTCGCCCAGGTTGTAAT
	R: GGAACTGATACTGGCACTCCT
GAPDH	F: GAAGGGTGGCGCCAAGAGGG
	R: GGGGGCCAAGCAGTTGGTGG
CTGF	F: CAAGGGCCTCTTCTGCGACT
	R: ACGTGCACTGGTATTTGCAG
IL-6	F: GCATTCCCTCCTCTGGTCA
	R: AAAACATTCAAGCCGCACA
COMP	F: GATGCGGACAAGGTGGTAGA
	R:TCCTGGTAGCCAAAGATGAAA
PRDX1	F:CCCAAGAAACAAGGAGGACTG
	R:GCCCCTGAATGAGATGCC
PRDX3	F:GAGCCCTGCATAACGAAGATG
	R:GAACTGGTGCTAAAGGCGAAT
PRDX4	F:TGATTCACAGTTCACCCATTTG
	R:CACGGGAAGGTCATTCAGAGTA
PRDX5	F:CCGTCGGTGGAGGTATTTG
	R:GGCAGGTGGGTCTTGGAAC
PRDX6	F:CTGGCAGGAACTTTGATGAGAT
	R:CCTCTTCAGGGATGGTTGGA

## Data Availability

Data are available upon request to the corresponding author.

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
