# Peer review of "PRDX2 Knockdown Inhibits Extracellular Matrix Synthesis of Chondrocytes by Inhibiting Wnt5a/YAP1/CTGF and Activating IL-6/JAK2/STAT3 Pathways in Deer Antler"

_ijms, 2022, doi:10.3390/ijms23095232_

Round 1

Reviewer 1 Report

“PRDX2 knockdown inhibits extracellular matrix synthesis of chondrocytes by inhibiting Wnt5a/YAP1/CTGF and activating IL-6/JAK2/STAT3 pathways in deer antler”

My comments are as follows:

Introduction

In the introduction, the advantages and rationale of using horn chondrocytes should be better explained.The deer is not considered an animal model.

Results

It would be useful to add images about antlers (also histology).

Before analyzing the expression of PRDX1-6 in antler chondrocytes, it would be useful to show chondrocytes characterization.

How did the authors distinguish prechondrocytes from chondrocytes? Again a characterization of these cells should be provided.

Figure 1B: what about other PRDXs?

Figure 1C: could the authors use a score to quantify cell positivity reporting a graph with statistical analysis?

Lines 110-112: the authors reported “PRDX2 110 knockdown significantly inhibited the expression of cartilage matrix proteins (Col2a, Sox9 and Aggrecan) compared with the negative control (Figure 2B)”. However, Sox9 is not reported in figure 2B.

Figures 4B and 4D: scale bar is missing. The cell images should be enlarged. It is difficult to see something.

To confirm the data obtained by siRNA, experiments with overexpression of PRDX2 should be performed or this point should added as a limitation of the study.

Figure 5: scale bar is not visible.

Section 2.5: the authors cited figure 4. I think figure 5 should be cited here.

Figure 5: after YAP1 knockdown, PRDX2 mRNA and protein expression should be evaluated.

Spaces between the captions of the figures and the text of the manuscript are necessary to avoid confusion in the reader.

Line 191: PRDX2 knockdown promotes the expression of IL6. Could the authors measure also protein levels? Are there other studies reporting that PRDX2 knockdown promotes the expression of IL6?

The authors reported that they performed three independent experiments. Did the authors use cells isolated from 3 different antlers in triplicate?

western blots in triplicate should be reported as supplementary materials. 

Discussion

Lines 248-250: the link between matrix calcification and this study should be better explained.

Line 264: “caspsae3” should be corrected.

The discussion should be improved. What are the consequences of this study? Are these data useful for human chondrocytes?

Limitations of the study are missing.

Methods

Could the authors specify the genus and specie of the deers used?

Line 281: “CO2” should be corrected.

Line 288: dilution used of PRDX2 antibody should be specified.

Section 3.4: did the authors use GAPDH as housekeeping gene?

Section 3.5: why did the authors block the membranes with skimmed milk? Skimmed milk should not be used when using phospho-antibodies.

Section 3.6-3.7: how many cells were seeded?

Abbreviations should be defined at first mention.

English should be improved.

Author Response

Thank you for your comments concerning our manuscript entitled:PRDX2 knockdown inhibits extracellular matrix synthesis of chondrocytes by inhibiting Wnt5a/YAP1/CTGF and activating IL-6/JAK2/STAT3 pathways in deer antler. Your comments are all valuable and very helpful for revising and improving our paper, as well as the important guiding significance to our research. For your comment, we have also made a detailed modification. At the same time, we also uploaded a word document.

Reviewer 2 Report

The authors studied the role of ROS scavenger PRDX2 gene during deer antler formation. They demonstrated a positive effect of this gene on chondrogenesis by promoting ECM synthesis and suppression of inflammation.

Comments

  1. Fig 1A: Housekeeping gene expression data and control gene expression levels should be demonstrated in both graphs. This should be corrected.
  2. Lines 106-115: Figs 2B and D are not described. Figure descriptions does not correspond to the text. This should be corrected.
  3. Line 112: The data on SOX9 expression is missing. This should be corrected.
  4. Line 162: The introduction of YAP1 is not clear. This should be corrected.
  5. Figs 1,2,4,5: Scale bars should be indicated on all pictures. This should be corrected.
  6. Lines 222-227: This text should be transferred to Introduction section.
  7. All the typos should be corrected.

Author Response

Thank you for your comments concerning our manuscript entitled:PRDX2 knockdown inhibits extracellular matrix synthesis of chondrocytes by inhibiting Wnt5a/YAP1/CTGF and activating IL-6/JAK2/STAT3 pathways in deer antler. Your comments are all valuable and very helpful for revising and improving our paper, as well as the important guiding significance to our research. For your comment, we have also made a detailed modification.

Fig 1A: Housekeeping gene expression data and control gene expression levels should be demonstrated in both graphs. This should be corrected.

Figure1A is just an electropherogram, we just want to prove that PRDXs family proteins are expressed in deer antler chondrocytes

Lines 106-115: Figs 2B and D are not described. Figure descriptions does not correspond to the text. This should be corrected.

    We have modified

Line 112: The data on SOX9 expression is missing. This should be corrected.

This is a mistake we made. Actually, what we want to describe is Col2a, Aggrecan and COMP (cartilage oligomeric matrix protein). Knockdown of PRDX2 did reduce Sox9 expression, but the protein was not part of the cartilage extracellular matrix. Therefore, we do not intend to present this result.

Line 162: The introduction of YAP1 is not clear. This should be corrected.

    We have modified

Figs 1,2,4,5: Scale bars should be indicated on all pictures. This should be corrected.

    We made the ruler clearer by zooming in on the picture

Lines 222-227: This text should be transferred to Introduction section.

    We have modified

All the typos should be corrected.

    We carefully checked the manuscript for typos

Round 2

Reviewer 1 Report

The manuscript improved after the revision. However, I have still some comments for the authors.

1) I asked to characterize prechondrocytes and chondrocytes to the authors. The authors replied that these cells were characterized in previous publications (I suppose of the authors). Could the authors add these data as supplementary file or cite these previous publications.

2) In the previous version, the authors reported that SOX9 was reduced after PRDX2 knockdown but the result was not shown. The authors replied that they did not want to show this result and deleted this part as SOX9 is not a cartilage extracellular matrix marker. I agree that SOX9 is not a marker of extracellular matrix but SOX9 is a pivotal gene in chondrocytes. Therefore, it would be important to understand the effect of PRDX2 knockdown. Thus, it should be discussed (at least) in the discussion as a future study etc.

2) Figure 1C: could the authors use a score to quantify cell positivity reporting a graph with statistical analysis? The authors replied that that is, there is a significant difference in the number of cells in the pre-cartilage layer and the cartilage layer. It is certain that the OD value of the cartilage layer in the same area is higher than that of the pre-cartilage layer through software analysis. I agree with the authors about this. However, they could normalize the OD value obtained taking into account the number of cells. Otherwise, it is possible to use semiquantitative grade such as: grade 0 = no stained cells, 1 = stained cells ≤ 20%, 2 = 21% < stained cells < 50%, 3 = 22% < stained cells < 50% etc adapting the grading.

3) figures 4 and 6: scale bar is not visible and the images are still not clear. Since there is no limit about figure number, I suggest to split the figure into two in order to enlarge the cell images.

4) The authors did not add experiments of PRDX2 overexpression. Thus, this point should be added as a limitation of the study in the discussion.

5) I requested to specify genus and specie of the deers used. The authors wrote “sica deer” in the methods, which corresponds to the scientific name Cervus nippon. Thus, Cervus nippon should be added in italics.

Author Response

Thank you very much for your comments, and we have carefully revised the manuscript in response to your comments. The following changes have been modified to your five suggestions.

(1)We cite our published articles in Results section 2.1 (Line 105-106)

(2) We follow your request that normalize the OD value obtained taking into account the number of cells (Figure.2D).

(3) In order to see the scale bar clearly, we have changed the color of the scale bar from red to white.

(4) We have added a description of this in the Discussion section, and this is a limitation of our study (Line 291-297)

(5) We have made changes in Materials and methods (Line 301)